# Partial Atrous Cascade R-CNN

**Mofan Cheng** [1] , **Cien Fan** [1,*], **Liqiong Chen** [1] , **Lian Zou** [1], **Jiale Wang** [1], **Yifeng Liu** [2] **and Hu Yu** [1]

[1] School of Electronic Information, Wuhan University, Wuhan 430072, China; chengmofan@whu.edu.cn (M.C.); liqiongchen@whu.edu.cn (L.C.); zoulian@whu.edu.cn (L.Z.); jialewang98@whu.edu.cn (J.W.); pacificationary@whu.edu.cn (H.Y.)

[2] National Engineering Laboratory for Risk Perception and Prevention (NEL-RPP), Beijing 100041, China; liuyifeng3@cetc.com.cn

[*] Correspondence: fce@whu.edu.cn

**Abstract:** Deep-learning-based segmentation methods have achieved excellent results. As two main tasks in computer vision, instance segmentation and semantic segmentation are closely related and mutually beneficial. Spatial context information from the semantic features can also improve the accuracy of instance segmentation. Inspired by this, we propose a novel instance segmentation framework named partial atrous cascade R-CNN (PAC), which effectively improves the accuracy of the segmentation boundary. The proposed network innovates in two aspects: (1) A semantic branch with a partial atrous spatial pyramid extraction (PASPE) module is proposed in this paper. The module consists of atrous convolution layers with multi-dilation rates. By expanding the receptive field of the convolutional layer, multi-scale semantic features are greatly enriched. Experiments shows that the new branch obtains more accurate segmentation contours. (2) The proposed mask quality (MQ) module scores the intersection over union (IoU) between the predicted mask and the ground truth mask. Benefiting from the modified mask quality score, the quality of the segmentation results is judged credibly. Our proposed network is trained and tested on the MS COCO dataset. Compared with the benchmark, it brings consistent and noticeable improvements in the case of using the same backbone.

**Keywords:** convolutional neural network; instance segmentation; partial atrous spatial pyramid extraction; mask quality

## 1. Introduction

Convolutional neural networks (CNNs) [1,2] have rapidly developed, leading to a series of methods in the fields of object detection [3], semantic segmentation [4], instance segmentation [5], etc. From the development of the state-of-art methods in these tasks, we can observe that some of these methods are generic and interacting. For example, excellent approaches [6–8] in object detection tasks provide precise bounding boxes, which are also widely used in instance segmentation tasks. Similarly, we try to use semantic features to achieve accurate and robust instance segmentation.

Instance segmentation emphasizes the semantic understanding of the scene. The image-level spatial context information in a semantic network promotes explicit pixel-wise prediction, which is also one of the core tasks of instance segmentation. In addition, the semantic segmentation task does not require high-level features to distinguish different instances, so the spatial size of its feature map is relatively large. The feature map from semantic segmentation can produce high-quality segmentation results, especially at the boundary of the segmentation. However, in most methods, the semantic features are underutilized. This is undoubtedly inappropriate for a task that needs to distinguish various objects at the pixel level. Roughly adding a semantic branch consisting of a simple combination of convolution layers is unreasonable and only brings limited gain in terms of mask average precision (AP) and box AP. An important reason for the large gap is the ignorance of spatial information. The methods roughly unify the pyramidal feature maps

to the same size, and use the same size of convolutional layer. This is bound to damage the multi-scale segmentation ability of the semantic branch.

To bridge this gap, we strive for a sophisticated design to extract as much spatial context information as possible. We note that the atrous convolution effectively improves the receptive field and propose partial atrous spatial pyramid extraction (PASPE), a module using atrous convolution with multi-dilation rates. Atrous convolution is a popular and efficient architecture, which increases the receptive field of the convolution kernel and avoids losing information in the process of pooling. On the basis of the encoder–decoder network, we redesign the entire semantic branch, which employs a PASPE module with the encoder–decoder structure. The features of the branch output are maintained at a high resolution, which supplements the loss of detail caused by the down-sampling in high-level features. The feature maps of the semantic branch are used in both the box branch and the mask branch. For the bounding box, the output feature is used to guide the distinction of the instances. For mask prediction, the pixel-level segmentation of the semantic branch encodes contextual information of the entire picture. The high-resolution feature maps especially benefit the boundary of the segmentation.

Meanwhile, we note the problem that the mask quality is a mismatch with the classification score. In most instance segmentation methods, classification score is used as the mask quality score directly. However, the box-level classification confidence is inappropriate to represent the pixel-level mask quality, especially when the object is partially blocked or overlaps with another object. As shown in Figure 1, the instance segmentation method hybrid task cascade (HTC) [9] infers classification and detection results with high confidence, but the corresponding segmentation results are unsatisfactory. The mismatch between the mask score and the actual mask quality not only leads to inaccurate supervision during training, but also results in suboptimal segmentation during inference. Considering the AP metric of the COCO dataset, we propose mask quality (MQ) module to calibrate the mask score. To unify the output and the evaluation metric, the MQ module is trained to regress the intersection over union (IoU) between the predicted mask from the mask head and the ground truth.

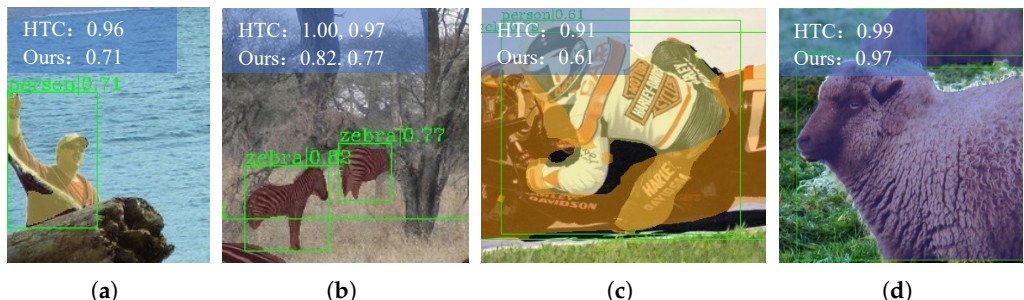

|  (a)  |  (b)  |  (c)  |  (d)  |

**Figure 1.** Score corrected by MQ module. For the mask with poor prediction results, such as part of the person's torso in (**a**), the zebra's limbs and head in (**b**) and the person in (**c**), HTC has a high score. Our score is lower but more reliable. For the case of a high quality prediction result, such as the sheep in (**d**), the MQ module maintains a high score.

The partial atrous cascade R-CNN (PAC) is easy to implement and can be trained in an end-to-end manner. To verify the superiority of our method, we trained and tested it on the MS COCO dataset [10]. The results show that it achieved 1.4% and 0.4% higher than the hybrid task cascade (HTC) [9] baseline in mask AP and box AP, respectively. The main advantages of our approach can be summarized as follows:

(1) We introduce the feature from the semantic branch into the instance segmentation task to exploit the accuracy improvement.
(2) We propose a partial atrous spatial pyramid extraction (PASPE) module that strengthens spatial context information and the boundary resolution. The semantic branch

integrated with the PASPE module accounts for discriminating objects from the cluttered background.

(3) We design a mask quality (MQ) module that calibrates the scores of masks. By regressing the IoU between the mask and the ground truth, the quality of the mask is correctly assessed. At the same time, appropriate mask quality evaluation improves segmentation performance.

## 2. Related Work

### 2.1. Instance Segmentation

Instance segmentation tasks are required to recognize objects of interest at the pixel-level. Generally speaking, the usual pipelines of instance segmentation can be categorized into two types: one is detection-based, the other is segmentation-based.

Detection-based methods pursue the goal of state-of-the-art performance. A major feature of this type of method is that region proposals need to be reported. The R-CNN [3] structure is widely applied in this regard. Unlike object detection tasks, instance segmentation requires instance discrimination at the pixel-level. Based on the target detection method Instance FCN [11], fully convolutional instance-aware semantic segmentation (FCIS) [12] fuses position-sensitive maps and inside/outside scores to perform end-to-end instance segmentation. Mask R-CNN [13] retains the basic framework of the detection method [8], and adds an mask branch to achieve mask prediction. RoI Align is proposed to improve the quantization accuracy of the feature maps. HTC [9] introduces the classic architecture of the cascade [14] into instance segmentation. HTC redesigns the hybrid cascade architecture to facilitate the information flow between multi-tasks. Mask Scoring R-CNN [15] points out the contradiction about the mask quality measurement standard. To solve the problem of the actual quality misjudgment, mask scoring is proposed. Refine-Mask [16] uses fine-grained features from the semantic branch to compensate for the loss of segmentation details. The boundary-aware refinement (BAR) strategy is also proposed to predict the boundary accurately. Look Closer to Segment Better [17] also focuses on boundary refinement. The patches extracted from the boundary are refined by boundary patch refinement (BPR). BCNet [18] considers the influence of occlusion between instances. A graph convolutional network (GCN) is used to decouple the boundaries of the occluding and occluded instances, and the interaction between the two is performed during mask prediction.

Segmentation-based methods balance performance and speed. The methods usually involve parallel processing of pixel-level category prediction and instance distinction. Since there is no need to regress the bounding box, the calculation is more efficient. YOLACT [19] refers the basic structure of the detection model RetinaNet [20]. The two modified parallel branches are used to generate prototype masks and mask coefficients of each instance. The subsequent version YOLACT++ [21] optimizes the mask evaluation criteria and convolution layer settings. PolarMask [22] contours based on the polar coordinate. Instance center classification and dense distance regression are used to complete the instance segmentation. Segmenting objects by locations (SOLO) [23] proposes an instance category, which meshes the input picture and sends it to different branches for category prediction and mask prediction.

### 2.2. Semantic Segmentation with Atrous Convolution

Semantic segmentation segments different types of targets without distinguishing instances. Modern semantic segmentation methods are developed from fully convolutional networks (FCNs) [24]. Many novel methods have been proposed to improve segmentation accuracy, such as the encoder–decoder architecture [25]. This approach proved to be effective in the medical field [26,27]. Atrous convolutions also bring considerable progress. The pooling layer is replaced by atrous convolution, which reduces information loss due to pooling. Hybrid dilated convolution (HDC) [28] proposes a design standard for atrous convolution, which solves the problems of the gridding effect and long-range/short-range information relevance. Atrous spatial pyramid pooling (ASPP) in DeepLab [29] processes

each scale with an independent branch. Multi-scale information extraction is achieved by using atrous convolutions of multiple dilation rates in a parallel manner. The method extracts multi-scale semantic information with a small amount of calculation. Subsequent improved versions such as DeepLab v3 [30] and v3+ [31] further optimize the ASPP module and network architecture, which take advantage of the full convolutional network to achieve efficient and accurate segmentation. The proposed shared decomposition convolution (SDC) and boundary reinforcement (BR) in [32] relieve the grid artifact problem and enhance the spatial information. Benefiting from its small amount of calculation and multi-scale information extraction ability, atrous convolutions are widely used in lightweight semantic segmentation, such as ENet [33] and ESPNet [34].

Unlike these methods considering instance segmentation and semantic segmentation independently, we are committed to feature supplementation and fusion. The features from our semantic branch are merged with the RoI feature, which brings spatial context and boundary information. The mismatch between mask score and mask quality is corrected by the additional MQ module. The network can be aware of the quality of the instance mask during training, and the mask score is more reliable during inference.

## 3. Method

### 3.1. Motivation

In this work, we propose partial atrous cascade R-CNN (PAC), a new framework of two-stage instance segmentation, as shown in Figure 2. The region proposal network (RPN) [3] is still used in the first stage, which proposes candidate instance bounding boxes. In the second stage, aiming at the absence of spatial context information, a new designed semantic branch is proposed to extract semantic features. As a supplement to image-level information, the outputs of the semantic branch contain a wealth of category-related semantic features. It is very helpful to distinguish segmentation targets. The proposed semantic branch is connected with the multi-task cascade branch in parallel. The information flow is designed to join each stage of the cascade branch, so that each stage can enjoy the benefits of semantic features. The multi-task cascade branch performs proposal classification, bounding box regression and mask prediction. In addition, in order to decouple the mask score with the bounding box score and independently evaluate the segmentation quality, an additional mask quality (MQ) module is integrated to judge the quality of the segmentation. The MQ module is arranged after the cascade branch, which revises the mask score using instance features and predicted masks. We present the details of the framework in the following sections.

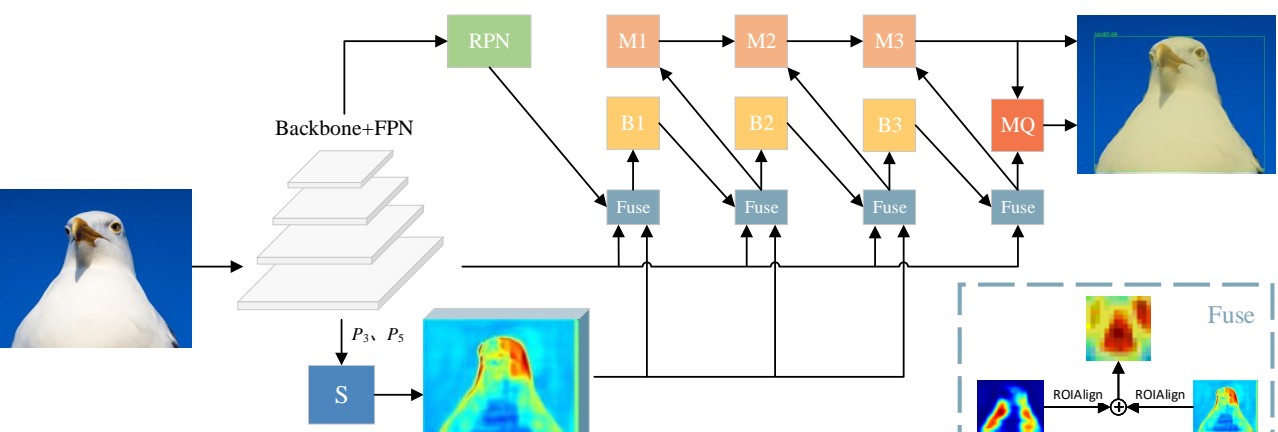

**Figure 2.** Network architecture of our method. S represents the semantic branch, Bn represents the n stage of the box branch, Mn represents the n stage of the mask branch and MQ represents the mask quality module.

### 3.2. Semantic Branch

The semantic branch aims to strengthen spatial context information and help the network to further distinguish foreground objects from background, especially the boundary regions where object confidence is hesitant. At the same time, benefiting from the large-scale semantic feature map, occluded pixels are also supplemented with information from other locations. Unlike the previous methods that unify feature maps and use the same kernel size convolution layers, our proposed branch introduces atrous convolution into the semantic branch.

The advantage of atrous convolution is that the size of the convolution kernel can be flexibly changed without increasing the amount of calculation. On the one hand, a large-size convolution kernel avoids the loss of spatial features caused by pooling. On the other hand, a flexible convolutional layer size captures spatial context features at a flexible scale. To make full use of that, we designed the partial atrous spatial pyramid extraction (PASPE) module with multiple dilation rates, atrous convolution layers and global pooling layers. As shown in Figure 3, the PASPE module consists of 5 convolution layers, including 3 atrous convolution layers and 2 normal convolution layers. The size of the atrous convolution layers is 3, the dilation rates are {6, 12, 18} and the output size is 256. Such large-sized and different-level dilation convolution kernels enrich semantic features. The sizes of the normal convolution layer are {1, 3} and the output size is 256. Such small-sized convolution kernels supply the local detailed features. The average pooling layer is equal to a convolution kernel of infinite size, which maintains global information. The features from the pooling layer are interpolated to the same size as others. In this way, the semantic branch not only has an excellent ability to extract global features, but also maintains the detailed information.

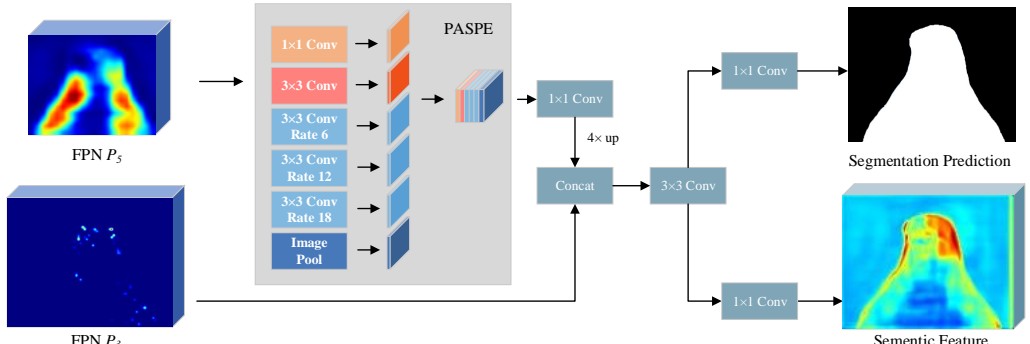

**Figure 3.** Architecture of the semantic branch. We introduce the PASPE module into the semantic branch to produce high-resolution feature maps that are rich in spatial context information.

The semantic branch in our network adopts the architecture of a fully convolutional network, and the main architecture is shown in Figure 3. Considering that the sixth stage feature map ($P_6$) of FPN is the down-sampling result of $P_5$, we choose $P_5$ as a high-level semantic feature input of PASPE. The outputs of PASPE are fused between channels by convolution layers with the kernel size of 1. The low-level features contain richer spatial details, which is precisely what high-level features lack. Therefore, the PASPE features are down-sampled and concatenated with the feature map from FPN $P_3$. There are two parallel convolution layers at the end of the semantic branch. One of them is responsible for semantic features. The features have the same channel as FPN to facilitate feature superposition (default 256), which is shared with other branches. The other channel performs semantic segmentation prediction and calculates the loss.

### 3.3. Mask Quality Module

The mask score and the classification score of the instance segmentation method usually share the same data. However, the box-level classification score irrelevantly represents pixel-level mask quality, especially when the instance is partially occluded by other objects.

As a result, a low-quality mask may maintain a high classification score. As shown in Figure 1, the conflict between high classification score and low mask quality is harmful to the task of segmentation that requires subdivision evaluation. During the training process, the network is inclined to prioritize classification over segmentation. In the inference process, the network selects the proposal with better classification results rather than better segmentation results. We propose a mask quality (MQ) module to unify the actual quality and the score of the mask. We can express the mask prediction task as $s_{score}$. As we mentioned earlier, $s_{score}$ regresses the IoU between the predicted mask from the mask head and the ground truth. Meanwhile, $s_{score}$ should remain positive for the ground truth category, and be close to 0 for other categories. As illustrated in Equation (1), we can divide the prediction $s_{score}$ into two subtasks: classify the mask, and regress the IoU between the mask and the ground truth. It is easy to obtain $s_{iou}$, which has been estimated in the classification branch. Furthermore, we utilize mask quality (MQ) to fit the IoU between the real mask and our evaluated one.

$$s_{score} = s_{cls} \cdot s_{iou} \tag{1}$$

The details of the MQ module are shown in Figure 4. The module concatenates feature maps from the fused features and predicted mask of M3. Only the dimension of the ground truth class of stage 3 (M3) is chosen. To obtain a uniform feature map space size, a max-pooling layer with the kernel size of 2 and stride of 2 is used to pool the M3 mask. The feature map is further processed by 4 convolutional layers and 3 fully connected layers. Figure 4 shows the parameters of the convolutional layers and the fully connected layers. The output of the final FC layer is set to the number of classes, so we adopt the ground-truth class of the output as $s_{iou}$. With $s_{score}$ and $s_{iou}$, $s_{mask}$ can be calculated by Equation (1).

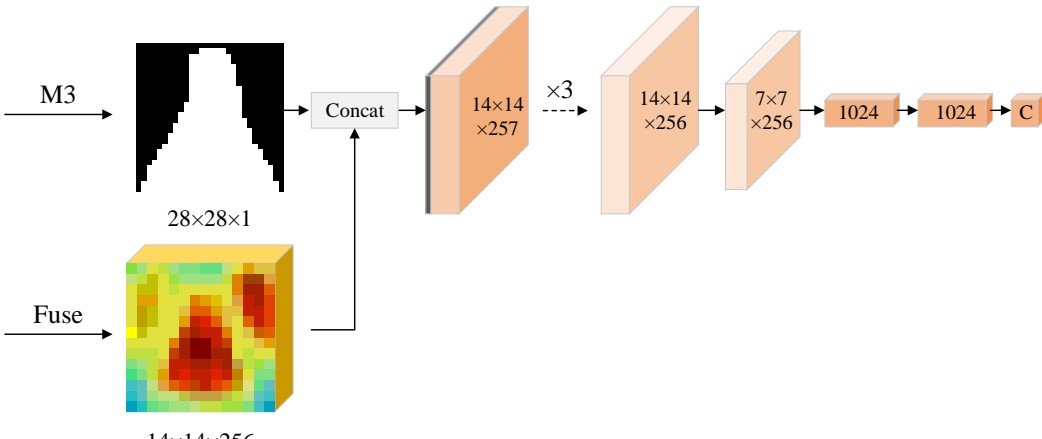

**Figure 4.** Architecture of mask quality (MQ) module. The parameters of all feature maps are marked in the figure. The feature map from M3 is max-pooled with the kernel size of 2.

### 3.4. Training Strategy

Our method can be trained in an end-to-end manner, and the overall loss function can be expressed in the form of multi-task learning as in Equation (2). Here, $L_{rpn}$ is the loss of the RPN, which is combined with two terms $L_{cls}$ and $L_{reg}$. As shown in Equation (3). The former classification loss adopts cross-entropy loss and the latter regression loss adopts smooth $L_1$ loss. $L_{bbox}^t$ is the loss of the bounding box predictions at stage $t$ of the box branch, which is also combined with $L_{cls}$ and $L_{reg}$, as shown in Equation (4). $L_{mask}^t$ is the loss of mask predictions at stage $t$, which adopts the binary cross-entropy loss. $L_{seg}$ is the semantic branch loss in the form of cross-entropy loss. $L_{mq}$ is the loss of the MQ module, which adopts $L_2$ loss. Hyperparameters $\alpha$, $\beta$ and $\gamma$ are used to balance the loss ratio of different

branches and stages. We follow the hyperparameter settings in HTC. By default, we set $\alpha = [1, 0.5, 0.25]$, $T = 3$, $\beta = 1$ and $\gamma = 1$.

$$L = L_{rpn} + \sum_{t=1}^{3} \alpha_t(L_{bbox}^t + L_{mask}^t) + \beta L_{seg} + \gamma L_{iou} \tag{2}$$

$$L_{rpn}(p, \hat{p}, t, \hat{t}) = L_{cls}(p, \hat{p}) + L_{reg}(t, \hat{t}) \tag{3}$$

$$L_{bbox}^t(c_t, \hat{c}_t, r_t, \hat{r}_t) = L_{cls}(c_t, \hat{c}_t) + L_{reg}(r_t, \hat{r}_t) \tag{4}$$

During inference, the feature maps from the backbone and semantic branch are intercepted by RoIAlign. They are concatenated as the input of the box branch and mask branch to obtain box results and mask results. The MQ module is used to calibrate the classification score generated from the box branch. We follow the standard HTC inference procedure, and the top-k masks are fed into the MQ module to predict $s_{iou}$. The calculated $s_{score}$ is used for instance selection.

## 4. Results

### 4.1. Datasets and Evaluation Metrics

Datasets: To verify the effectiveness of the method, we performed experiments on the MS COCO dataset. We followed COCO settings of the 2017 version with 80 object categories. A total of 115 k training images were used to train the model, 5 k validation images were used for validation and 20 k test images were used for the test. Typical instance annotations were used to supervise box and mask branches, and the semantic branch was supervised by stuff annotations.

Evaluation Metrics: We use the average precision (AP) metrics of COCO-style to report the result, which averages APs using IoU thresholds from 0.05 to 0.95 with an interval of 0.05. For the mask, we report $AP_{50}$, $AP_{75}$, and $AP_S$, $AP_M$, $AP_L$.

### 4.2. Implementation Details

We maintained a 3-stage cascade box and mask branch. We chose ResNet-18 [35] for ablation studies, and ResNet-50/ ReNeXt-101 [36] for comparing our method with other baseline results. FPN was used in all backbones. Our method was implemented with PyTorch and mmdetection. We trained detectors with 4 GPUs. The initial learning rate was 0.02 in the case of the 16 batch size and was linearly adjusted according to the change in batch size. Our model was trained for 20 epochs. The learning rate was decreased by 1/10 after 16 and 19 epochs. The input images were resized to 1333 pixels for the long edge and 800 pixels for the short edge. Synchronized SGD was used while optimizing. As in HTC, bounding boxes were refined progressively by box branches of different stages during inference. The boxes that scored higher than a threshold (0.001 by default) were used for the mask branch and MQ module.

### 4.3. Semantic Branch Optimization Results

During the entire semantic branch design, we tried to keep the branch efficient and considerable. The proposed semantic branch fuses feature maps of multi-levels. Here we specifically output semantic segmentation results to comparatively analyze the optimization effect. As shown in Figure 5, since the semantic branch ultimately contributes to object detection and instance segmentation, we can focus on the instance segmentation quality in the image. It can be seen from the comparison of the same group of pictures that the segmentation quality of different types of instance has been improved. Especially for the boundary part, the segmentation becomes clear and firm. Such performance serves for more credible detection of bounding boxes and more accurate foreground segmentation.

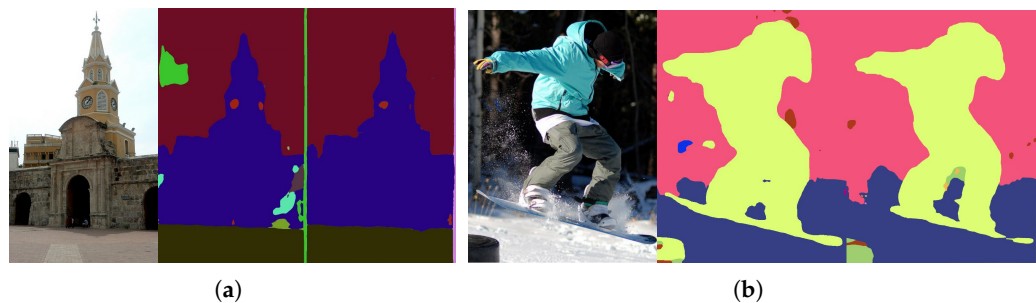

<div align="center">(<b>a</b>)          (<b>b</b>)</div>

**Figure 5.** Semantic branch optimization results. The first picture of each group is the original picture, the second picture is a semantic prediction of HTC and the third picture is a semantic prediction of our method. Comparing the right part of the building in (**a**) and the legs of the skier in (**b**), the accuracy is improved in our method.

### 4.4. MQ Module Optimization Results

Benefiting from the MQ module, the score of the prediction mask has a strong relationship with mask quality. The MQ module effectively corrects the problem that the actual quality of the mask mismatches with the classification score. We choose experimental results here to verify the effect. As shown in Figure 1, for the same mask of the object, the scores before and after correction have been marked. It can be seen that for the mask with poor prediction results, HTC still predicts a high score while the score corrected by the MQ module is more reliable. The error between score and quality is calibrated. For the mask with an appropriate prediction result, the corrected score is still high. The score reflects the actual quality of the mask. The design correctly guides the optimization orientation of the training process and filters out the mask of low quality during inference.

### 4.5. Quantitative Results

We compared our method with different backbone networks including ResNet-18/50 and ResNeXt-101. The results are shown in Table 1. We used mask AP to represent instance segmentation results and box AP to represent object detection results. The experiment was divided into 3 parts: network with our proposed semantic branch, with network MQ module and network with both optimizations above. As can be seen from the results of the semantic head group, our proposed semantic branch has an effect on both the box branch and the mask branch. The box AP is increased by 0.5% and the mask AP is increased by 0.2%. It can be seen from the result that the MQ module has an optimization effect on the mask branch, and that the mask AP increases by about 0.7%. Our method deploys both optimizations at the same time. We treated HTC as the benchmark. From the result in Table 2, our proposed semantic branch and MQ module significantly improves the performance to 39.5% (1.4% improvement) and 41.5% (1.2% improvement) with backbone ResNet-50 FPN and ResNeXt-101 FPN, respectively. The visualization of the segmentation results of our method is shown in Figure 6.

**Table 1.** Experimental results for MS COCO. The object detection and instance segmentation results are reported, corresponding to the box AP and mask AP in the table. The results without any ✓ are those of HTC, with ✓ under MQ being those that add the MQ module, with ✓ under SH being those that apply our proposed semantic branch, and with ✓ under MQ and SH at the same time being those of our method PAC.

| Backbone | SH | MQ | Mask AP | $AP_{50}$ | $AP_{75}$ | Box AP | $AP_{50}$ | $AP_{75}$ |
|---|---|---|---|---|---|---|---|---|
| ResNet-18 FPN | | | 34.0 | 53.8 | 36.6 | 38.3 | 56.2 | 41.5 |
| | ✓ | | 34.5 | 54.9 | 36.8 | 38.8 | 57.4 | 41.9 |
| | | ✓ | 35.0 | 53.7 | 37.6 | 38.3 | 55.5 | 41.6 |
| ResNet-50 FPN | | | 38.1 | 59.4 | 41.0 | 43.2 | 62.1 | 46.8 |
| | ✓ | | 38.3 | 59.8 | 41.2 | 43.7 | 62.8 | 47.3 |
| | | ✓ | 38.9 | 58.9 | 42.0 | 43.0 | 60.8 | 46.7 |
| | ✓ | ✓ | 39.5 | 60.3 | 42.6 | 43.6 | 61.9 | 47.2 |
| ResNeXt-101 FPN | | | 40.3 | 62.2 | 43.5 | 46.1 | 65.3 | 50.1 |
| | ✓ | | 40.3 | 62.5 | 43.6 | 46.2 | 65.6 | 50.2 |
| | | ✓ | 41.1 | 62.1 | 44.5 | 45.8 | 63.8 | 49.8 |
| | ✓ | ✓ | 41.5 | 62.6 | 44.9 | 46.2 | 65.4 | 50.2 |

**Table 2.** Comparison with other instance segmentation methods for MS COCO dataset.

| Method | Backbone | Box AP | Mask AP | $AP_{50}$ | $AP_{75}$ | $AP_S$ | $AP_M$ | $AP_L$ |
|---|---|---|---|---|---|---|---|---|
| Mask R-CNN [13] | ResNet-50 FPN | 39.2 | 35.4 | 56.4 | 37.9 | 19.1 | 38.6 | 48.4 |
| Mask R-CNN [13] | ResNeXt-101 FPN | 42.2 | 37.8 | 59.6 | 40.6 | 19.8 | 41.4 | 51.9 |
| MS R-CNN [15] | ResNet-50 FPN | 38.8 | 36.3 | 56.1 | 39.2 | 18.8 | 39.3 | 50.8 |
| MS R-CNN [15] | ResNeXt-101 FPN | 41.8 | 38.7 | 59.3 | 41.9 | 20.8 | 42.3 | 52.9 |
| BPR [17] | ResNeXt-101 FPN | - | 39.2 | - | - | - | - | - |
| BCNet [18] | ResNet-50 FPN | - | 38.4 | 59.6 | 41.5 | 21.9 | 40.9 | 49.3 |
| BCNet [18] | ResNet-101 FPN | - | 39.8 | 61.5 | 43.1 | 22.7 | 42.4 | 51.1 |
| HTC [9] | ResNet-50 FPN | 43.2 | 38.1 | 59.4 | 41.0 | 20.3 | 41.1 | 52.8 |
| HTC [9] | ResNeXt-101 FPN | 46.1 | 40.3 | 62.2 | 43.5 | 22.3 | 43.7 | 55.5 |
| RefineMask [16] | ResNet-50 FPN | - | 38.2 | - | - | - | - | - |
| RefineMask [16] | ResNeXt-101 FPN | - | 41.0 | - | - | - | - | - |
| PAC | ResNet-50 FPN | **43.6** | **39.5** | 60.3 | 42.6 | 21.1 | 42.8 | 55.0 |
| PAC | ResNeXt-101 FPN | **46.2** | **41.5** | 62.6 | 44.9 | 23.1 | 45.3 | 57.3 |

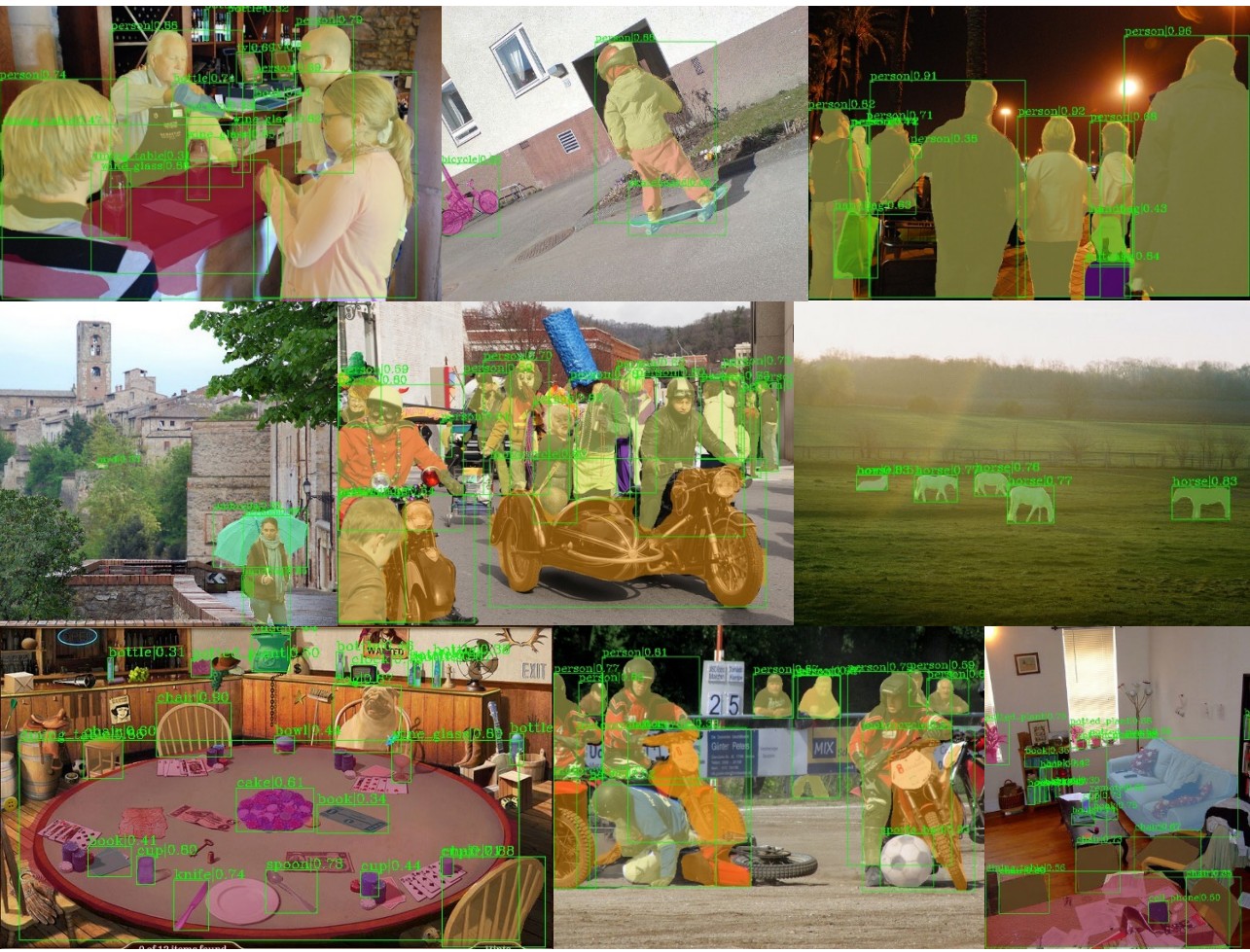

**Figure 6.** Examples of instance segmentation results for our method.

## 5. Ablation Study

In this part, ResNet-18 with FPN was used for all ablation studies. The model was trained for 12 epochs.

### 5.1. The Architecture of the MQ Module

We studied the design of the MQ module input and the convolution layer setting. The design choices are shown in Figure 7 and explained as follows.

Mask information flow: Considering that mask information flow performs well in the mask branch, we try to introduce the mask information flow into the MQ module. Following similar principles, we introduce an information flow between the mask branch and MQ module by feeding the mask features of mask branch stage 2 to the MQ module. The design is shown in Figure 7a.

Architecture simplification: Considering that a large number of convolution layers in the mask branch have played a feature extraction role, we appropriately reduce the number of convolution layers in the MQ module to reduce the amount of calculation. The design is shown in Figure 7b.

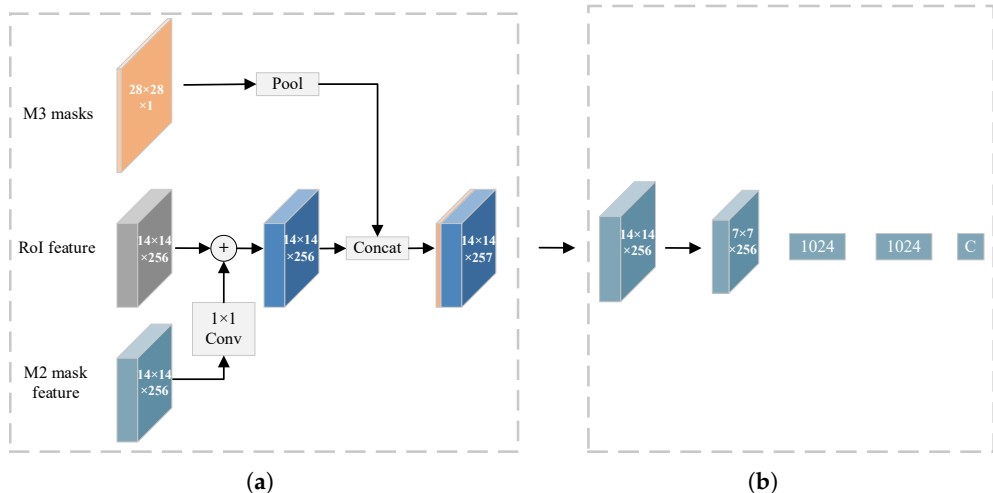

(**a**)                                              (**b**)

**Figure 7.** Different design of MQ module architecture: (**a**) shows the different design of the input, (**b**) shows the simplified architecture.

The results are shown in Table 3. We focus on the first four lines. Compared with the benchmark, we can see that all the versions of the MQ module design can bring performance improvements in mask AP, and the original MQ module design has the most significant improvement. Thus, we take it as the default design.

### 5.2. The Architecture of the PASPE Module

The atrous convolution of the ASPP module takes into account instances of different scales. However, we found that the module has low segmentation accuracy for small targets because of lacking small-size convolution kernels. We chose to add a redesigned semantic head and PASPE module to enhance the feature extraction capability for the instance boundary. The details of the design choice are shown in Figure 8.

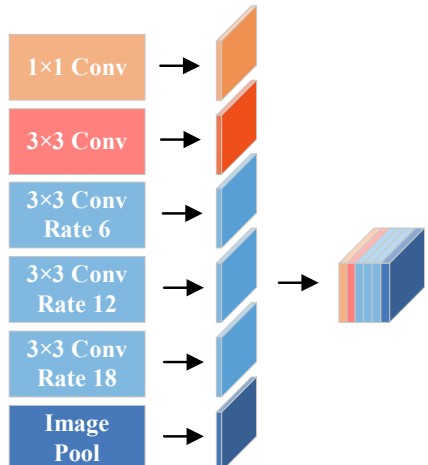

**Figure 8.** PASPE module architecture.

The results are shown in Table 3. We focus on the last two lines. Compared with the ASPP module, our proposed PASPE module design can bring more accurate results in both box AP and mask AP.

**Table 3.** Results for different design of MQ module and PASPE module architecture.

| Design | Box AP | Mask AP | AP$_{50}$ | AP$_{75}$ |
|---|---|---|---|---|
| HTC baseline [9] | 38.3 | 34.0 | 53.8 | 36.6 |
| +MQ module | 38.3 | 35.0 | 53.7 | 37.6 |
| +Mask information flow | 38.4 | 34.5 | 53.5 | 37.0 |
| +Architecture simplification | 38.2 | 34.9 | 53.7 | 37.7 |
| +ASPP [31] | 38.8 | 34.5 | 54.9 | 36.8 |
| +PASPE | 38.9 | 34.6 | 55.0 | 37.0 |

## 6. Conclusions

We have proposed a high-quality instance segmentation framework called partial atrous cascade R-CNN (PAC) in this paper. The image-level spatial context information from semantic features is introduced into instance segmentation in our method. PAC adopts a newly designed semantic branch that utilizes the PASPE module, which accurately distinguishes different instances and corrects the boundary of the mask. The branch improves both box and mask prediction. To solve the mask quality misjudgment problem, a mask quality (MQ) module is introduced to calibrate the mask score. Without bells and whistles, extensive results show that PAC consistently and obviously outperforms the benchmark on the MS COCO dataset. The proposed semantic branch and MQ module can also be applied to other instance segmentation methods to improve performance.

**Author Contributions:** Conceptualization, M.C. and Y.L.; methodology, M.C., C.F. and L.Z.; software, M.C. and J.W.; validation, M.C. and J.W.; formal analysis, L.C.; investigation, L.C.; resources, L.Z. and C.F.; data curation, M.C. and Y.L.; writing—original draft preparation, M.C. and H.Y.; writing—review and editing, M.C., C.F. and H.Y.; visualization, M.C.; supervision, L.C.; project administration, C.F. and L.Z.; funding acquisition, Y.L. All authors have read and agreed to the published version of the manuscript.

**Funding:** This research received no external funding.

**Institutional Review Board Statement:** Not applicable.

**Informed Consent Statement:** Not applicable.

**Data Availability Statement:** Not applicable.

**Acknowledgments:** The numerical calculations in this paper were performed on the supercomputing system in the Supercomputing Center of Wuhan University.

**Conflicts of Interest:** The authors declare no conflict of interest.

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
