# Peer review of "Partial Atrous Cascade R-CNN"

_electronics, doi:10.3390/electronics11081241_

Round 1

Reviewer 1 Report

The paper proposes a partial atrous spatial pyramid extraction module that strengthens spatial context information and the boundary resolution. The proposed framework includes a two-stage instance segmentation. In the first stage, the RPN structure is used for proposing candidate instance bounding boxes. In the second stage, a new semantic branch outputs semantic features by extracting spatial context information from features of backbone. The proposed method is evaluated on MS COCO dataset, and it consistently performs better than prior methods on the ResNet-50 FPN and ResNeXt-101 FPN backbones. 

The paper researches the instance segmentation problem, it is a significant problem in deep learning and the proposed method performs well in this problem. For my opinion, it is a good work in general. However, the authors should have a better clarification on the novelty of the work in section 3 (Method). For my understanding, there is some novelty in the network architecture design of the framework, but I am not sure if the novelty is significant enough. Also, in section 3.4 the authors discuss the loss function used in the model training. Does the work use the same loss function as the prior work or include some novelty on the loss design? 

Author Response

Thank you for acknowledging the work of this paper. 

We modified the first paragraph of section 3 (Method) , and describe the motivation for improvement and design innovation in more detail.

In the task of instance segmentation, most methods use similar loss functions. For the part without structural modification, the loss functions follow the previous methods. For the part we proposed, such as semantic branch, we add a loss for supervised semantic features, which uses L2 Loss.

Reviewer 2 Report

The authors proposed a new instance segmentation method by applying the partial atrous spatial pyramid extraction (PASPE) to the semantic branch of the R-CNN model. They also proposed a new score of Mask Quality (MQ) which utilizes a intersection over union regressd on the mask. Results are MS COCO dataset is promising and ablation study is included.

Overall, the proposed method is well-defined, and experiments are comprehensive. However, there are quite a few writing issues that needs to be improved.

First, the full names of the abbreviations should be spelled in full when first mentioned in the text (e.g. FCIS, HTC, HDC). 

Second, as the mask scoring is an important element of this paper. The author should motivate this better by including more details about traditional methods. For instance, the description in line 91-92 is too little, and if the authors include more details, and potentially add an additional figure that illustrates the contradiction about the mask quality measurement, the paragraph would read much better.

In the Methods section, there are many "Optimization Result". The referee believes that those should belong to the Results section instead of the Methods section. Please restructure the article properly.

Author Response

Thank you for acknowledging the work of this paper.

For the first comment, we recheck the paper, and spell the abbreviations in full when it is first mentioned. 

For the second comment, we modify the fourth paragraph of section 1 (Introduction) and supply more details about mask quality judgement.  We describe the motivation and the flaws of previous methods, and cite figure to illustrate it.

For the last comment, we move the "Optimization Result" to the section 4 (Results).